# **Reference Floating Wind Array Designs for Three Representative Regions**

Leah H. Sirkis<sup>1</sup>, Ericka Lozon<sup>1</sup>, and Matthew Hall<sup>1</sup>

<sup>1</sup>National Renewable Energy Laboratory, 15013 Denver West Parkway, Golden, CO 80401

Correspondence: Leah H. Sirkis (leah.sirkis@nrel.gov)

**Abstract.** This work presents the systematic development of three open-source reference floating wind array designs. The designs are tailored to representative site conditions for three regions of the United States: Humboldt Bay off the coast of California, the Gulf of Maine, and the Gulf of America. We adopted existing reference designs for the individual 15 MW turbines, semisubmersible floating platforms, substations, mooring systems, and power cables - integrating and adapting them as needed for each location. We adapted existing dynamic cable designs to use larger conductor sizes to meet the arrays' power transmission requirements, and we set up redundant mooring systems for each substation. The layout of each array is a uniform grid design optimized to approximately minimize the levelized cost of energy (LCOE) within a square lease area while satisfying spatial constraints. These constraints ensure adequate clearances between adjacent turbines and between underwater components during the layout optimization to prevent clashing and ensure that all components reside within the lease boundaries. Substations are included to allow accounting for intra-array cable costs. They are placed within the uniform grid to maintain the navigability of the arrays. For each feasible layout considered, annual energy production and cable routing costs are calculated and updated in the LCOE objective function. After the optimization, we refined the cable routing with a mix of algorithmic and manual methods to ensure that the cables avoid mooring system components and approach the substation with adequate clearances. We confirmed the suitability of each reference array's layout by comparing the wake losses at each wind heading angle to the wind rose, observing that the optimized layouts largely avoid wake losses in the predominant wind directions. These reference arrays provide open-source baseline designs to enable future research and innovation of floating wind technology at the array scale.

#### 1 Introduction

Floating wind turbines can access strong and consistent wind resources while also positioning wind farms farther from shore, reducing visual impacts and conflicts with other ocean co-users; however, floating wind is still a developing technology, and there are no existing large-scale floating wind arrays. The largest floating wind array, Hywind Tampen, has 11 turbines with a

https://doi.org/10.5194/wes-2025-209 Preprint. Discussion started: 20 October 2025 © Author(s) 2025. CC BY 4.0 License.

25

45

combined capacity of 94.6 MW, whereas fixed-bottom wind farms have advanced to gigawatts of capacity. Floating wind array design has only recently become an area of significant research.

Reference designs – open-source definitions of representative systems – have helped floating wind research and development by giving researchers a common starting point and baseline for comparison. Research at the single turbine level has produced various reference designs at increasing sizes as turbine technology advances over time. The earliest widely used examples are the National Renewable Energy Laboratory (NREL) 5 MW reference wind turbine (Jonkman et al., 2009) and the Offshore Code Comparison Collaboration (OC3)-Hywind spar-buoy reference platform (Jonkman, 2010). Another widely used reference floating platform developed for the 5 MW turbine is the Offshore Code Comparison Collaboration Continuation (OC4)-DeepCWind Semisubmersible (Robertson et al., 2014). Both floating system reference designs also include definitions of the tower, control system, and catenary chain mooring system. Reference turbine capacity increased with the Technical University of Denmark 10 MW reference wind turbine (Bak et al., 2013), which was used in research on a range of floating platforms, including public semisubmersible designs in the LIFES50+ project (Yu et al., 2018). The INO WINDMOOR base case reference wind turbine system (Silva de Souza et al., 2021) – including a turbine, semisubmersible floating platform, and mooring system – was developed with a capacity of 12 MW.

The most widely used reference floating system at present is the International Energy Agency (IEA) Wind Technology Collaboration Programme (IEA Wind) 15 MW reference wind turbine (Gaertner et al., 2020) and the University of Maine (UMaine) VolturnUS-S semisubmersible floating platform (Allen et al., 2020). Several other floating platform designs were developed to work with the IEA Wind 15 MW reference turbine as well, including the Windcrete spar and the ActiveFloat semisubmersible reference designs (Mahfouz, 2020). All three of these support structure reference designs were developed with a chain catenary mooring system. In recent years, larger reference wind turbines have been developed, such as the IEA Wind 22 MW reference turbine (Zahle et al., 2024), which includes a semisubmersible design based on the UMaine VolturnUS-S semisubmersible but is tailored to fit the 22 MW turbine.

In general, considerations for underwater components – such as moorings, dynamic cables, and anchors – were limited in the aforementioned reference systems. The mooring systems were basic catenary designs with uniform lengths of chain. Anchors and dynamic power cables were rarely specified. More complete underwater component reference designs have been developed in recent years. Janocha et al. (2024) developed reference power cable design definitions for floating wind systems, including a set of reference cable properties. Lozon et al. (2025) designed mooring and dynamic power cable reference designs for shallow, moderate, and deep water for three representative locations in the United States, including catenary, semitaut, and taut mooring configurations; however, reference definitions of floating wind arrays consisting of multiple floating wind turbines and their associated underwater components have not yet been published.

To aid floating wind research at the array level, there is a need for reference floating array designs comprising mooring systems, dynamic power cables, static cable routing, and the full layout of these items in the array. Reference designs serve future research by providing a baseline and starting point for further exploration.

Previous studies have developed several fixed-bottom reference wind farm designs. The Norwegian Research Centre for Offshore Wind Technology reference wind farm developed a 1.2 GW fixed-bottom uniform grid array design based on the

https://doi.org/10.5194/wes-2025-209 Preprint. Discussion started: 20 October 2025 © Author(s) 2025. CC BY 4.0 License.

Dogger Bank Creyke Beck A sizing and location (Kirkeby and Tande, 2014). The Norwegian Research Centre for Offshore Wind Technology reference farm included cable routing and a study on the use of 33 kV versus 66 kV collector systems. The Norwegian Centre for Offshore Wind Energy developed a uniform grid and an irregular (non-gridded) 0.8 GW reference fixed-bottom wind farm for conditions in the North Sea (Bak et al., 2017), including cable layouts, operations and maintenance activities, and cost analyses. The IEA Wind Task 55 project developed a set of reference fixed-bottom wind farm arrays based on the Borssele III and IV lease areas off the coast of Belgium and the Netherlands, where they optimized a uniform grid array layout and an irregular array layout (Kainz et al., 2024). The IEA Wind Task 55 reference farm included cable routing and conductor sizing, and the layout optimization accounted for the water depth of the site. There are currently no floating open-source reference wind array designs to the authors' knowledge, representing a significant gap in floating wind research.

To develop floating wind reference arrays, layout optimization methodologies specific to floating systems are needed. Floating farm layouts require a wide variety of considerations to ensure a feasible and holistic design, including the design and constraints of components that are specific to floating wind, array layout optimization, and intra-array cable routing. Considering all these factors in a floating wind farm layout optimization represents a significant challenge. Mooring systems for floating wind farms – which often have a large, site-specific footprint – must fully reside within the lease area boundaries; therefore, floating array layout optimization must consider the spatial constraints for mooring design and orientation. Varied bathymetry and sediment in the array can also affect the design of specific mooring lines, anchors, and dynamic cables, which can affect the overall costs and mooring footprints. Moorings, platforms, and cables also must not clash with each other. Further considerations for navigability, installation, operations and maintenance, and supply chain availability can also factor into the feasibility of a floating wind layout.

Floating wind layout optimization techniques require an optimization algorithm, an objective function, and floating-specific constraints. There are a variety of optimization algorithms, and research has not yet converged on a specific algorithm to best optimize wind farm layouts. A comparison of optimization algorithms for fixed-bottom wind farm layouts revealed that various different techniques produced similar levelized costs of energy (LCOE) (Thomas et al., 2023).

The development of constraints and objective functions for floating wind layout optimization has been approached with a variety of priorities and considerations using a wide range of optimization algorithms. Lerch et al. (2021) used particle swarm optimization to optimize the electrical layout of a floating wind farm for LCOE. Eikrem et al. (2023) used an ensemble optimization method, a form of stochastic optimization that uses an ensemble of controls to approximate a gradient, which is often used in oil reservoir optimization problems. Though ensemble methods often cannot handle constraints, they include them by breaking the problem into subproblems that apply penalty functions to optimize a floating wind farm layout for LCOE and annual energy production (AEP) using a minimum spanning tree algorithm to determine the intra-array cable layout. Rapha (2023) developed an optimization technique for floating wind layouts that algorithmically adjusted the moorings and cables based on bathymetry to account for their changing spatial footprints and costs. Mahfouz et al. (2024) considered the mooring design for wake steering in the layout optimization process. Heitanen et al. (2024) developed a layout optimization tool that included a binary anchor choice based on the soil type and added buffer zones along the mooring lines. The tool maximizes the net present value with a random search optimization algorithm, specifically modeling costs that are affected by the layout of

https://doi.org/10.5194/wes-2025-209

Preprint. Discussion started: 20 October 2025

© Author(s) 2025. CC BY 4.0 License.

105

the array. Hall et al. (2024a) developed a layout optimization approach that included anchor selection based on the soil type and mooring adjustment for bathymetry. They used a sequential least-squares gradient-based optimization algorithm. This tool was further developed by Sirkis et al. (2025) to add intra-array cable routing and sizing with a minimum spanning tree algorithm as well as anchor sizing. They used a particle swarm optimization and a sequential least-squares gradient-based optimization algorithm.

In this paper, we develop reference floating wind array designs for three regions in the United States: Humboldt Bay, the Gulf of Maine, and the Gulf of America. We directly use the mooring and dynamic cable designs developed in Lozon et al. (2025) for these same regions in complete gigawatt-scale array designs. We approximately optimize each array layout with an approach that builds on the layout optimization tool developed in Hall et al. (2024a) and Sirkis et al. (2025) to include novel cable routing techniques and improved layout optimization methods. These array designs will serve as some of the first open-source reference floating array designs, with fully shared design details to facilitate future use and application. The full definition files for these designs are available on GitHub at https://github.com/FloatingArrayDesign/ReferenceDesigns.

The layout of this paper is as follows: Section 2 describes the general array design process methodology; Sect. 3 defines the component and array designs for Humboldt Bay, the Gulf of Maine, and the Gulf of America, respectively; and, finally, Sect. 4 describes conclusions and future work.

#### 2 Array design methodology

The floating wind array design methodology presented in this paper builds on the techniques in Hall et al. (2024a) and Sirkis et al. (2025) to include improvements to the layout optimization approach and cable routing techniques. We applied the same array design methodology to the development of a reference array design for each of three U.S. regions. The overall reference array design process, shown in Fig. 1, can be described in five general steps:

- 1. Site parameter selection
- 2. Component type selection
- 3. Component design optimization
  - 4. Layout optimization
  - 5. Cable routing adjustment.

We based the site parameter selection on the three reference site condition sets developed in Biglu et al. (2024a) based in Humboldt Bay, the Gulf of Maine, and the Gulf of America. These reference site definitions include meteorological ocean (metocean) characteristics for extreme and fatigue load analysis as well as bathymetry and soil type information representative of several U.S. regions. Component type selection and design optimization were completed in Lozon et al. (2025). That work used reference metocean, depth, and soil parameters based on Biglu et al. (2024a). The depth and soil information informed

Figure 1. Array design process overview

the anchor type, the mooring configuration, and the cable type selections. These components were then optimized to minimize cost and meet critical constraints for these conditions.

The present work builds on the site conditions and component designs from previous work to create full floating wind array designs. We apply the mooring and dynamic cable designs within a layout optimization to create layouts of the wind turbines, the mooring lines, and the power cables that minimize LCOE considering the spatial dependencies from wind rose data, lease area boundaries, and required array cables. After the layout is optimized, we adjust the chosen layout's cable routing to prevent clashes between moorings, cables, and anchors. This completes the reference floating wind array design for each region.

To serve as general reference designs, each site's characteristics are simplified to use a uniform seabed and square lease area, and the export cable to shore is not included.

#### 2.1 Site parameter selection

We considered site-specific bathymetry, soil types, and metocean conditions when developing the reference array designs. These data are based on Biglu et al. (2024b), which pulled metocean data for each site from several sources – including the National Data Buoy Center, NREL's National Offshore Wind Dataset, and the High Frequency Radar Network – and then processed the data to extrapolate extremes at different return periods and fatigue bins. These data were used in Lozon et al. (2025) to define the critical load cases for designing mooring systems and dynamic power cables for the three sites. We used wind roses developed for each site from the same site condition dataset (Biglu et al., 2024b) during the array layout optimization process.

The reference array designs are meant to be representative of the general region they were designed for rather than fitting in a specific lease area; therefore, we assume that all lease areas have a constant bathymetry and soil type representative of the region and that the lease boundaries are square. The area within the lease boundary is based on the size of the lease areas in each region.

## 2.2 Subsystem and component design

The reference arrays use reference component and subsystem designs developed in previous work when available and relevant to the needs of each site. The following subsections detail the design selections and the nature of any design adaptations.

# 2.2.1 Floating wind turbine

The floating wind turbine assumed for the reference array designs is the IEA Wind 15 MW reference turbine (Gaertner et al., 2020). It is a widely used reference wind turbine design, developed through a collaborative effort as part of the IEA Wind Task 37 on Wind Energy Systems Engineering. The platform is the University of Maine VolturnUS-S reference semisubmersible (Allen et al., 2020), which was specifically designed for the IEA Wind 15 MW turbine. The VolturnUS-S is a generic steel semisubmersible with three radial columns and a central column that holds the tower. The platform and wind turbine are shown in Fig. 2, and their properties are summarized in Table 1. The VolturnUS-S platform with the IEA Wind 15 MW turbine provide a well-established floating wind turbine system for the reference array designs.

The VolturnUS-S was originally designed with a chain catenary mooring system for a 200 m depth. We replaced the mooring system with designs from Lozon et al. (2025) to suit the water depths of the reference array designs. Section 2.2.3 discusses the mooring designs in more detail. We also added dynamic power cables (Sect. 2.2.5), which were not included with the original VolturnUS-S design.

Table 1. VolturnUS-S and IEA 15 MW reference wind turbine properties

| Parameter                           | Value  |
|-------------------------------------|--------|
| Turbine rating (MW)                 | 15     |
| Hub height (m)                      | 150    |
| Rotor diameter (m)                  | 240    |
| Rated wind speed (m/s)              | 10.59  |
| Freeboard (m)                       | 15     |
| Draft (m)                           | 20     |
| Platform mass (t)                   | 17,839 |
| Tower mass (t)                      | 1,263  |
| RNA mass (t)                        | 991    |
| Hull displacement (m <sup>3</sup> ) | 20,206 |

## 2.2.2 Floating substation

The floating offshore substation design used in these arrays is a rectangular semisubmersible high-voltage alternating current (HVAC) substation developed by Jorge Alcantara (2023). This design has a capacity of 1.2 GW. The platform comprises four

Figure 2. RAFT model of VolturnUS-S semisubmersible and IEA Wind 15 MW wind turbine

square columns connected in a square with four pontoons. The dimensions and mass properties of the floating substation platform are shown in Table 2. The geometry of the floating substation platform is visualized in Fig. 3.

As with the floating wind turbine, we applied the mooring line and dynamic cable designs developed in Lozon et al. (2025) to the floating substation; however the substations feature a larger number of mooring lines to ensure redundancy of the design. A substation failure would have a more significant impact on farm revenue than a single turbine failure, so a redundant mooring system is of greater importance. We verified the performance of each mooring system and substation under 500 year return period extreme wind, wave, and current loading in the open-source frequency domain modeling tool RAFT (Hall et al., 2022) to ensure acceptable platform offsets for the dynamic cable designs. All substations feature intra-array cable connections on a maximum of three of the four sides; one side is free of cables to allow maintenance vessel access.

**Table 2.** Floating offshore substation design developed by Jorge Alcantara (2023)

| Parameter                                             | Value |
|-------------------------------------------------------|-------|
| Platform length (m)                                   | 54.78 |
| Platform width (m)                                    | 54.78 |
| Cable deck height above mean water level (m)          | 12.00 |
| Draft (m)                                             | 22.00 |
| Mass (mT)                                             | 29084 |
| Vertical center of gravity below mean water level (m) | 5.63  |

Figure 3. RAFT model of floating substation

https://doi.org/10.5194/wes-2025-209 Preprint. Discussion started: 20 October 2025

# 2.2.3 Mooring system

The mooring systems in the reference array designs were previously developed in Lozon et al. (2025) for the same sets of site conditions. These designs each have a different configuration (catenary, semitaut, and taut in order of increasing depth). They were optimized for the extreme and fatigue site conditions in a multifidelity modeling process consisting of (1) the optimization of line dimensions to minimize costs subject to initial constraints checked in the quasi-static mooring model MoorPy (Hall, 2024), (2) extreme and fatigue load analyses using the dynamic floating wind turbine modeling tool OpenFAST (Jonkman et al., 2023), and (3) adjusting tuning factors in the quasi-static optimization and iterating until all constraints were satisfied. The design constraints included maximum tensions, fatigue life of chain sections, avoiding polyester rope contact with the seabed, yaw stability, avoiding vertical loading on drag-embedment anchors, and platform offset. This design process and the resulting designs are detailed further in Lozon et al. (2025).

We used these mooring system designs directly for the floating wind turbines in the reference array designs, relying on the extreme and fatigue load analyses and constraint checks that were performed in Lozon et al. (2025); however, for the floating substations, we used the same mooring line designs but increased the number of mooring lines to six or eight to provide increased restoring stiffness and redundancy. We verified that the substation mooring systems keep the platform offsets within acceptable limits under extreme current loading.

#### 2.2.4 Anchors

Detailed anchor design was not a focus of this work, but anchor costs significantly contribute to the overall array cost; therefore, approximate anchor masses were directly pulled from Lozon et al. (2025), which sized anchors based on maximum anchor loads from extreme load cases performed in OpenFAST and general soil types. We input these approximate anchor masses into our anchor cost modeling assumptions described in Sect. 2.3.3 to estimate the anchor material costs.

# 2.2.5 Power cables

The reference array designs include intra-array cables between the turbines and from the turbines to the substation. Each intra-array cable connecting two floating wind turbines or a floating wind turbine and a substation consists of a dynamic cable on either end to connect to the platform and a static cable routed along the seabed between the dynamic cables.

The dynamic cable designs were developed in Lozon et al. (2025) following a similar design process as the moorings. That work initially optimized the cable dimensions and checked constraints in MoorPy, and then it checked the dynamic cables in OpenFAST against constraints for extreme tensions and allowable curvature in extreme load cases, iterating until all constraints were met. More details on the design process of the dynamic cables can be found in Lozon et al. (2025).

For the gigawatt-scale reference floating wind farms, additional dynamic cable designs were needed for larger conductor sizes. These larger conductor sizes are necessary to meet the power transmission needs when many turbines are connected in series within the wind farm. We adapted the dimensions of the initial designs from Lozon et al. (2025), which use cable with a 300 mm<sup>2</sup> conductor cross-sectional area, for larger conductor sizes (630 and 1000 mm<sup>2</sup>) by increasing the number of buoyancy

modules to compensate for the increased cable weight. This approach maintains approximately the same cable profile shapes and ranges of motion. We then simulated these additional cable designs in conjunction with the floating wind turbine and mooring system in OpenFAST to ensure compliance with the allowable cable tensions and curvatures.

The properties for each dynamic cable are shown in Table 3. For all dynamic cable designs, we assumed a buoyancy module with a volume of 0.57 m<sup>3</sup>, consistent with the original reference designs. The buoyancy module properties are shown in Table 4.

**Table 3.** Dynamic cable properties

| Parameter                             | Cable Type 1 | Cable Type 2 | Cable Type 3 |
|---------------------------------------|--------------|--------------|--------------|
| Conductor size (mm <sup>2</sup> )     | 300          | 630          | 1000         |
| Outer diameter (m)                    | 0.161        | 0.184        | 0.203        |
| Linear density (kg/m)                 | 36.66        | 55.76        | 75.74        |
| Axial stiffness (MNm <sup>2</sup> )   | 469          | 658          | 854          |
| Bending stiffness (kNm <sup>2</sup> ) | 19.92        | 42.47        | 68.73        |
| Min. bearing load (kN)                | 383.2        | 537.4        | 698.4        |
| Min. bending radius (m)               | 2.41         | 2.76         | 3.05         |

Table 4. Buoyancy module properties

| Parameter                            | Value  |
|--------------------------------------|--------|
| Displaced volume (m <sup>3</sup> )   | 0.566  |
| Mass (kg)                            | 270.68 |
| Overall density (kg/m <sup>3</sup> ) | 500    |
| Length (m)                           | 0.90   |
| Diameter (m)                         | 0.865  |
|                                      |        |

The static cables are represented by their routing along the seabed and their cross-sectional properties. Cable burial and environmental loadings are beyond the reference design scope. The properties for each static cable are shown in Table 5.

#### 2.3 Layout

To develop the reference array layouts, we used and expanded on an NREL-developed layout optimization tool described in Hall et al. (2024a) and Sirkis et al. (2025) to minimize the LCOE. This tool considers moorings, anchors, and cables, and it is capable of interfacing with a variety of optimizers. Figure 4 summarizes the array layout design process, which is described in the following subsections.

Table 5. Static cable properties

| Parameter                             | Cable Type 1 | Cable Type 2 | Cable Type 3 |
|---------------------------------------|--------------|--------------|--------------|
| Conductor size (mm <sup>2</sup> )     | 300          | 630          | 1000         |
| Outer diameter (m)                    | 0.154        | 0.177        | 0.197        |
| Linear density (kg/m)                 | 30.18        | 45.33        | 60.87        |
| Axial stiffness (MNm <sup>2</sup> )   | 287          | 417          | 551          |
| Bending stiffness (kNm <sup>2</sup> ) | 7.68         | 17.60        | 29.59        |
| Min. bearing load (kN)                | 183.7        | 260.5        | 342.8        |
| Min. bending radius (m)               | 2.31         | 2.66         | 2.95         |

Figure 4. Array layout design process

# 2.3.1 Layout design parameterization

Each reference array layout follows a uniform grid approach with seven key design variables that control the grid geometry. We chose a uniform grid to maintain navigability within the array, following the U.S. Coast Guard recommendations (United States Department of Homeland Security and United States Coast Guard, 2024). The design variables we used for these uniform grid layout optimizations are as follows:

- $D'_x$ ,  $D'_y$ : Grid x, y spacing (m)
- $x_0$ ,  $y_0$ : Grid x, y translation from centroid (m)
- $\alpha$ : Grid rotation (°)
- $\beta$ : Grid skew (°)
- $\gamma$ : Platform rotation (°)

The grid variables are shown in Fig. 5. The platform rotation variable,  $\gamma$ , is defined as 0° when one platform leg (or mooring line) is due north. Platform rotation definitions are independent of the grid rotation angle  $\alpha$ . All angles are measured clockwise positive.

Figure 5. Grid design variables, adapted from Hall et al. (2024b)

The substation rotation is defined at 0° when one pontoon faces each cardinal direction. We set the substation heading after the optimization process based on the spatial requirements of the array as well as the direction of the incoming dynamic cable strings.

We altered the uniform grid layout optimization methodology in Sirkis et al. (2025) to improve the computational efficiency and give more consistent results. The previous method tested each potential platform location against spatial constraints before adding that grid location to the layout, and it stopped adding points when the required number of platforms was met. In the current method, we develop a grid of all possible platform location points inside the lease area without determining if the constraints are met. In cases where more grid points than the required number of platforms fit inside the boundary, points closest to the boundary are removed until the required number of platforms remain. This ensures that the array layout is approximately centered within the lease area.

Platforms closest to the boundary are generally most at risk of violating spatial constraints, such as mooring system components crossing the boundary; therefore, when generating the layouts, we keep the platform locations with the best chance of passing spatial constraint checks without checking the constraints of every possible grid point. Layout constraints are checked all together at a later step for the grid platforms to improve efficiency. When less than the required number of platforms fit in the boundary, the layout is excluded from consideration.

# 2.3.2 Substation placement and preliminary cable routing

45 Cable routing within the array is dependent on the location of the substation(s), and the intra-array cables are an important contributor to cost; therefore, it is important to accurately represent the substation placement during the layout optimization.

To maintain navigability of the array, we assume that substations must be positioned on the same uniform grid as the turbines. The approach in Sirkis et al. (2025) kept the substation location constant during the optimization process, which does

https://doi.org/10.5194/wes-2025-209 Preprint. Discussion started: 20 October 2025 © Author(s) 2025. CC BY 4.0 License.

not allow the substation location to be part of the grid. Our current approach places substations in the uniform grid at the grid points closest to the user-inputted substation positions. This allows the user to choose the general substation locations while ensuring that the substations fit within the layout's uniform grid. The total number of grid points maintained when developing the platform locations, as described in Sect. 2.3.1, includes the total number of turbines plus the number of substations to accommodate substations in the grid.

Once the substation positions are defined, an approximate cable routing is automatically performed within the optimization loop. This cable routing allows for an approximation of the cable costs during the optimization. We then refine the cable routing after the optimization is finished, as discussed in Sect. 2.3.6.

When there are multiple substations in the array, we first assign each turbine to a substation before determining the cable routing. This differs from Sirkis et al. (2025), which only supported one substation. To support multiple substations, we use an assignment algorithm that allocates turbines to their closest substation to reduce cable costs. If the number of turbines connected to a substation exceeds the substation's capacity, turbines that have the smallest difference in distance to an alternate substation from the overwhelmed substation are re-allocated to the alternate substation until each is at or below capacity.

After each turbine is assigned to a substation, we apply the cable routing approach described in Sirkis et al. (2025) for the pool of turbines assigned to each substation. First, the clusters of turbines to be connected in series are determined using spectral clustering around the substation. Then, the routing within the clusters is determined using Prim's algorithm (Prim, 1957), a minimum spanning tree method. Sirkis et al. (2025) describes this intra-array cable routing algorithm in detail. The conductor size for each cable is determined based on the power requirements from the number of upstream turbines, which affects the cost, as described in Sect. 2.3.3.

# 2.3.3 Optimization objective function

The objective of the layout optimizations is to minimize the LCOE. To improve computational efficiency, the optimization framework only calculates the LCOE for layouts that meet all constraints described in Sect. 2.3.4. The LCOE can be described as:

$$LCOE = \frac{FCR \times CapEx + OpEx}{AEP},\tag{1}$$

where FCR is the fixed charge rate, defined as the fraction of capital expenditure (CapEx) that will be paid each year; CapEx is the total capital expenditure for the array, including the installation and component costs; OpEx is the annual operational expenditure; and AEP is the annual energy production.

The array AEP is calculated using the Gaussian curl hybrid wake model within the steady-state wake modeling tool FLORIS (v4.2) (Gebraad et al., 2014). The wind roses we used in the AEP calculations cover each direction at intervals of 5° for every wind speed at intervals of 1 m/s.

For the layout optimization, all CapEx costs except for those of the intra-array cables are assumed constant throughout the optimization. The intra-array cable material costs are updated for each feasible layout considered in the optimization based on the output cable routing. During the optimization, these cable costs are approximated by the 2-dimensional distance between

turbines multiplied by the dynamic cable cost per meter. This simplification is used to improve computational efficiency. It results in shorter cable lengths than a three-dimensional representation; however, the cost reduction is partially offset by the higher dynamic cable cost in comparison with static cables. The cable cost per unit length for a 66 kV dynamic cable is modeled as:

$$Cost_{iac} = (\$0.7845/\text{mm}^2/\text{m})A + \$257.3/\text{m},$$
 (2)

where  $Cost_{iac}$  is the dynamic intra-array cable cost per unit length, and A is the cable conductor area. The dynamic cable cost values are based on Hall et al. (2024b).

The mooring material cost is calculated based on the material cost per unit length as follows:

$$Cost_{chain} = (-\$312/\text{m}^2)d + (\$8.56 \times 10^4/\text{m}^3)d^2,$$
(3)

$$Cost_{poly} = (\$117/\text{m}^2)d + (\$1.27 \times 10^4/\text{m}^3)d^2, \tag{4}$$

where d is the diameter,  $Cost_{chain}$  is the mooring chain cost per unit length, and  $Cost_{poly}$  is the mooring polyester cost per unit length. Cost coefficients are in 2024 U.S. dollars. These cost values are based on the data and assumptions provided in Davies et al. (2025).

The anchor material costs are determined based on the material cost per kg as follows:

$$Cost_{DEA} = (\$4.150/\text{kg})m,\tag{5}$$

$$Cost_{SPA} = (\$4.435/\text{kg})m,\tag{6}$$

where  $Cost_{DEA}$  is the material cost of the drag-embedment anchors,  $Cost_{SPA}$  is the material cost of the suction pile anchors, and m is the anchor mass. The drag-embedment anchor material cost coefficient is based on the data and assumptions in Davies et al. (2025). The suction pile material cost coefficient is based on the average cost per mass value provided in Hall et al. (2024b). Because of the constant bathymetry in the arrays, one mooring and anchor design is used for all platforms in a given array, so the mooring and anchor cost remains constant for each array optimization.

The remaining CapEx costs and the OpEx costs are based on the data and assumptions provided in Housner and Mulas Hernando (2024). The CapEx costs excluding mooring, cable, and anchor materials are calculated at a rate of \$3,749/kW of capacity. Annual OpEx costs are calculated at a rate of \$62.5/kW of capacity. The FCR is set at 5.82 %.

After the optimization, we implement realistic three-dimensional cable designs with lazy-wave cable configurations at each platform connected to static cable sections along the seabed, as described in Sect. 2.2.5. Additionally, we refine the cable routing around the substation and to avoid moorings and anchors, which is described in Sect. 2.3.6. The additional component cost calculations used in the three-dimensional representation of the cables are described in the following.

Buoyancy module costs are calculated as:

$$Cost_{buoy} = (\$8590/m^3)V + \$3080,$$
 (7)

where  $Cost_{buoy}$  is the buoyancy module cost, and V is the buoyancy module volume. These values were determined from industry estimates.

Figure 6. Anchor, mooring, and platform buffer zones top-down view

Static cable costs are calculated as:

$$Cost_{static} = (\$0.719/\text{mm}^2/\text{m})A + \$239.57/\text{m},$$
 (8)

where  $Cost_{static}$  is the cost per unit length, and A is the cable conductor area.

The estimated costs of the cable connectors, which include bend stiffeners, are as follows for each cable:

$$Cost_{connectors} = (\$212.22/\text{mm}^2)A + \$139831,$$
 (9)

where  $Cost_{connectors}$  is the cable connector cost per cable.

Cable joints, found at the transition between the dynamic and static cable sections, are modeled as a constant cost of  $$237 \times 10^{3}$  per turbine. The static cable, cable connectors, and cable joint costs are based on the data provided in Hall et al. (2024b).

#### 2.3.4 Spatial constraints

325

330

Spatial constraints are checked during the array layout optimization to ensure realistic and feasible designs. The spatial constraints apply buffer zones around the mooring lines, anchors, and platforms to ensure that these components do not cross each other and stay within the boundaries of the lease. Fig. 6 shows the buffer zones that are applied around a single floating wind turbine.

We use the same approach to buffer zones as laid out in Hall et al. (2024a). Anchor buffer zones have a 100 m diameter centered around the anchor, which ensures that no two anchors are less than 100 m apart, per ISO (2005). The mooring buffer zones have a 40 m diameter centered along the axis of the mooring line. Mooring buffer zones may not cross anchor buffer zones or other mooring buffer zones, and anchor buffer zones may not cross other anchor buffer zones. Platforms also have a

https://doi.org/10.5194/wes-2025-209 Preprint. Discussion started: 20 October 2025 © Author(s) 2025. CC BY 4.0 License.

buffer zone with a 400 m diameter. The spacing between turbines is set to a lower limit of 0.6 nautical miles or 1111 m, but the platform buffer zone ensures a minimum distance from the lease boundary edge. Mooring line and anchor buffer zones may cross the platform buffer zone. To keep the design within the lease area boundaries, no buffer areas are permitted to cross a boundary.

We do not check if cables cross mooring lines or other components in the optimization process because the cable routing developed in the optimization is designed to estimate cable costs by simply determining the shortest distance between connected platforms rather than determining the exact route of a cable between two platforms; therefore, cables do not have buffer zones within the optimization. This simplification is used to improve the computational efficiency of the optimization. After the optimization is completed, a full 3-dimensional representation of the cables is implemented. The dynamic cable headings and the static cable routing points are then adjusted to avoid mooring and anchor clashing, as described in Sect. 2.3.6. As discussed in Sect. 2.3.3, the cost difference is limited and does not greatly affect the total.

# 2.3.5 Optimization approach

345

350

355

360

The layout optimization, where the grid parameters are adjusted to minimize the LCOE while meeting spatial constraints, can be done with many types of optimizers. We chose a particle swarm optimizer for its ability to find the global minimum even when there are discontinuities and many local minima. A particle swarm optimizer is a gradient-free optimization method developed by Kennedy and Eberhart (1995), based on the natural phenomenon of animals' collective behavior in a swarm, such as schooling fish. An initial randomized group (swarm) of particles, each representing a potential solution in the design space, is evaluated, and each particle moves within the design space at each iteration. Each particle considers its best known solution as well as the swarm's best known solution. This method requires many function evaluations per iteration, but it allows the optimizer to move past local minima. Note that the goal of this work is to develop and present detailed, open-source array layout designs that approximately minimize the LCOE while meeting the constraints and design requirements of each region; therefore, a detailed study of the optimization algorithms and settings that lead to the global minimum LCOE is out of scope.

# 2.3.6 Post-optimization cable routing and adjustment

After the layout optimization is completed, we refine the preliminary intra-array cable routing with an algorithmic approach that identifies and adjusts cables that are at risk of clashing with mooring lines. We apply an angular buffer on all mooring lines along the mooring line heading, and we examine if a dynamic cable heading lies within the angular buffer zones of the platform it is attached to. A 30° angle is used by default, but we adjust this value to fit the unique spatial requirements of each array. The angular buffer begins at the center of the platform and extends for 500 m past the cable attachment point on the platform. Cables that cross the buffer are adjusted to follow the outside of the angular buffer for 500 m from the cable attachment point or for the horizontal span of the dynamic cable, whichever is longer.

After this distance, cables begin routing toward the next turbine, even if the mooring radius is larger than 500 m, because the angular buffer increases the distance from the mooring with length. Continuing the angular buffer for the entirety of the

**Figure 7.** Cables reroute to avoid angular mooring buffers (not to scale)

Figure 8. Cables reroute around anchor buffer zones (not to scale)

mooring line length could cause the cable to interfere with the moorings of other turbines for locations with large mooring footprints, such as Humboldt Bay. Figure 7 shows this process.

If a static cable overlaps with an anchor buffer zone, we reroute the cable around the anchor with an additional routing point placed 100 m from the anchor point in a direction perpendicular to the initial cable heading, as shown in Fig. 8.

We also apply some manual routing adjustments to cables at the substation. Cables attaching to a substation are rerouted to ensure that one side of the substation is clear of cables and to avoid acute angles when possible for the static cable routing. The headings of the dynamic cables entering the substation are spaced at 5° intervals to prevent clashing between dynamic cables.

#### 370 3 Reference array designs

We developed reference array designs for reference site conditions representative of three regions: Humboldt Bay, the Gulf of Maine, and the Gulf of America. Each region distinctly varies in water depth and metocean conditions. Following the methodology outlined in Sect. 2, we applied the mooring system and dynamic cable designs and developed optimized array layouts and cable routing for each region.

The Humboldt Bay and Gulf of America reference arrays feature 67 turbines for approximately 1 GW of installed capacity, while the Gulf of Maine reference array features 132 turbines for approximately 2 GW of installed capacity. The Gulf of Maine array is larger to match the capacities of the proposed lease areas in that region.

A summary of the design characteristics for each region is shown in Table 6.

Table 6. Summary of design characteristics for each region

| Parameter              | Humboldt Bay | Gulf of Maine  | Gulf of America |
|------------------------|--------------|----------------|-----------------|
| Number of turbines     | 67           | 132            | 67              |
| Array capacity (MW)    | 1005         | 1980           | 1005            |
| Total lease area (km²) | 256          | 504.5          | 280             |
| Number of substations  | 1            | 2              | 1               |
| Water depth            | 800          | 200            | 80              |
| Mooring type           | Taut         | Semitaut       | Catenary        |
| Cable type             | Lazy wave    | Lazy wave      | Lazy wave       |
| Anchor type            | Suction pile | Drag-embedment | Drag-embedment  |

The following subsections further describe each reference array design.

#### 380 3.1 Humboldt Bay

The Humboldt Bay array design uses taut mooring systems, which are suitable for the deep-water depth of 800 m, and lazy-wave dynamic cables. The array consists of 67 turbines, resulting in a capacity of 1.005 GW. There is a single substation, located near the center of the array, with nine cable strings. We chose this location to reduce the required length of the large-conductor-size intra-array cables. The array design was challenged by large anchoring radii for the mooring systems and long dynamic cable spans, which required more careful positioning of elements within the full wind farm to maintain the necessary clearances.

Figure 9. Humboldt Bay (a) wind, (b) wave, and (c) current roses (Biglu et al., 2024a)

# 3.1.1 Site conditions

Humboldt Bay is located off the coast of California. The water depths in the Humboldt Bay lease areas range from 550 to 1100 m (Cooperman et al., 2022), with a uniform 800 m reference depth assumed for the array design. We selected a square lease area of 256 km<sup>2</sup> based on the size of the Humboldt Bay northeast lease area.

The Humboldt Bay area has large extreme current speeds, ranging from 0.92 to 1.44 m/s. The wind rose is mostly unidirectional, with the wind coming predominantly from the north. The wind, wave, and current roses for the Humboldt Bay reference site conditions are shown in Fig. 9. The extreme load case conditions, including the design load cases (DLC) 1.6, 6.1, and a survival load case (SLC), are shown in Table 7.

Table 7. Humboldt Bay extreme load case conditions

| Parameter            | DLC 1.6 | DLC 6.1 | SLC   |
|----------------------|---------|---------|-------|
| $H_S$ (m)            | 10.5    | 11.8    | 13.7  |
| $T_P(m)$             | 18.7    | 19.8    | 21.4  |
| Current speed (m/s)  | 0.92    | 1.09    | 1.44  |
| Wind speed (m/s)     | 10.59   | 39.44   | 42.97 |
| Turbulence intensity | .06     | .05     | .05   |

Figure 10. Humboldt Bay mooring and dynamic cable system (Lozon et al., 2025)

# 3.1.2 Mooring and cable design

The Humboldt Bay mooring design, developed in Lozon et al. (2025), is taut with suction pile anchors. In this work, we assume lazy-wave cables throughout the array. Figure 10 shows the Humboldt Bay mooring and dynamic cable configuration.

The Humboldt Bay mooring design is taut, consisting mostly of polyester rope with chain sections at the anchor and fairlead connections. The anchoring radius of the mooring system is 1400 m, which is significantly larger than the Gulf of Maine and Gulf of America designs. The mooring design is summarized in Table 8. Further details on the Humboldt mooring design performance can be found in Lozon et al. (2025).

The Humboldt Bay dynamic cable configuration is a lazy wave. The initial dynamic cable design for a 300 mm<sup>2</sup> cable was optimized by Lozon et al. (2025) with a cable span of 800 m (the horizontal distance between the platform connection and the joint or transition point to the static cable) and a buoyancy section length of 400 m. When evaluating the cable routing within the full array, we found that the large cable span made it difficult to fit mooring lines and cables without crossing. To address

**Table 8.** Humboldt Bay mooring line design adapted from Lozon et al. (2025)

| Parameter                 | Value                    |
|---------------------------|--------------------------|
| Anchoring radius (m)      | 1400                     |
| Fairlead radius (m)       | 58                       |
| Fairlead depth (m)        | 14                       |
| Pretension (kN)           | 1704                     |
| Declination angle (°)     | 36.6                     |
| Line section 1 material   | 120 mm R4 studless chain |
| Line section 1 length (m) | 80                       |
| Line section 2 material   | 184 mm polyester         |
| Line section 2 length (m) | 1378.9                   |
| Line section 3 material   | 120 mm R4 studless chain |
| Line section 3 length (m) | 80                       |

this, we decreased the cable span from 800 m to 500 m while keeping the other cable dimensions the same. This effectively reduced the length of the dynamic cable that is always laying on the seabed. The dynamic cable designs for the 630 and 1000 mm<sup>2</sup> conductor areas have the same span, total cable length, buoyancy section length, and buoyancy section midpoint location; however, we optimized the number of buoyancy modules, and consequently the buoyancy module spacing, for each design. The 300, 630, and 1000 mm<sup>2</sup> designs require 34, 50, and 74 buoyancy modules, respectively. The buoyancy module spacing ranges from 12.1 m to 5.5 m. The three dynamic cable designs are summarized in Table 9.

# 3.1.3 Optimized layout

We optimized the Humboldt Bay array layout to maximize the LCOE. The parameters of the optimized array design are listed in Table 10. The *x* and *y* spacing are 1847.2 and 1431.3 m, respectively, and there is a small amount of skew. The grid is rotated 36.7° maximizing spacing in the predominant wind direction of due north. The substation is located in the center of the array. The Humboldt Bay mooring system has a large anchoring radius of 1400 m, which required careful positioning of the mooring systems to fit 67 turbines within the area. As a result, the turbine rows alternate between two opposite mooring orientations to fit the turbines more closely together. To avoid interference with mooring lines that run along the columns, we used an angular buffer of 30° to reroute the lazy-wave cables away from the mooring line heading, as described in Sect. 2.3.6. In some locations, the static cable is routed beneath the mooring lines. This maintains acceptable clearances because the taut mooring system is mostly suspended. The array layout and cable routing are shown in Fig. 11. The static cables were automatically rerouted to avoid intersecting with anchors, as shown in Fig. 12. The rerouting follows the logic outlined in Fig. 8.

Table 9. Dynamic power cable design parameters for Humboldt Bay

| Parameter                                  | Value   |         |         |
|--------------------------------------------|---------|---------|---------|
| Conductor size (mm <sup>2</sup> )          | 300     | 630     | 1000    |
| Cable span (m)                             | 500     | 500     | 500     |
| Fairlead radius (m)                        | 5       | 5       | 5       |
| Total cable length (m)                     | 1070.43 | 1070.43 | 1070.43 |
| Length of cable below buoyancy section     | 297.98  | 297.98  | 297.98  |
| Midpoint of buoyancy section (m)           | 497.98  | 497.81  | 497.98  |
| Buoyancy section length (m)                | 400     | 400     | 400     |
| Length of cable above buoyancy section (m) | 372.45  | 372.45  | 372.45  |
| Number of buoyancy modules                 | 34      | 60      | 89      |
| Buoyancy module spacing (m)                | 12.07   | 6.76    | 4.52    |
| Averaged diameter of buoyancy section (m)  | 0.290   | 0.377   | 0.451   |
| Averaged mass of buoyancy section (kg/m)   | 59.17   | 96.63   | 136.5   |

The Humboldt Bay substation design features six mooring lines, with two corners supported by two mooring lines and the opposite corners supported by one mooring line each. Though it would be preferable to have two mooring lines on each corner for improved symmetry, we implemented a six-line design for Humboldt Bay due to spatial constraints. This mooring design adheres to the maximum allowable offsets dictated by the dynamic cable designs when modeled under extreme current loading. To provide sufficient clearances around this mooring system, we rerouted the dynamic cables to two sides of the substation with headings 5° apart, as shown in Fig. 13.

Table 10. Humboldt Bay reference array layout design variables

| Parameter                       | Value     |
|---------------------------------|-----------|
| Grid $x$ spacing, $D'_x$ (m)    | 1847.2    |
| Grid $y$ spacing, $D_y'$ (m)    | 1431.3    |
| Grid $x$ translation, $x_0$ (m) | -494.8    |
| Grid $y$ translation, $y_0$ (m) | -3552.0   |
| Grid rotation, $\alpha$ (°)     | 36.7      |
| Grid skew, $\beta$ (°)          | 7.3       |
| Platform rotation, $\gamma$ (°) | 3.1, 63.1 |

We designed the Humboldt Bay array layout to avoid the predominant wind direction of north-south, as shown by the wake plot in Fig. 14a. Figure 14b shows the wake losses for the array with a 12 m/s wind speed for every wind heading at 1° intervals.

Figure 11. Humboldt Bay array layout and cable routing in (a) plan view and (b) 3 dimensions

The wake losses are at a maximum of approximately 30 % when the wind is oriented along the columns (i.e., northwest to southeast). The wake losses are slightly less along the rows because the spacing is larger. The wind rose shows that the wind is predominantly coming from the north to northwest directions, which have minimal wake losses. This shows that the Humboldt Bay array layout was well designed to minimize wake effects.

The final values affecting the LCOE calculations in the optimization process are described in Table 11. These cost values are based on the cost curves in Sect. 2.3.3 and reflect the final design, which includes the refined cable routing. The cable material costs and mooring system material costs are similar, while the anchor material costs are substantially less.

Table 11. Humboldt Bay reference array layout AEP and mooring, cable, and anchor CapEx

| Performance Metric        | Value |
|---------------------------|-------|
| AEP (GWh)                 | 4873  |
| Cable CapEx (\$M)         | 202.2 |
| Mooring lines CapEx (\$M) | 168.3 |
| Anchor CapEx (\$M)        | 66.1  |

Figure 12. Humboldt Bay array cable rerouting to avoid anchors

#### 3.2 Gulf of Maine

The Gulf of Maine array design features semitaut mooring systems and lazy-wave dynamic cables. It has 132 turbines for a total capacity of 1.98 GW. Two substations are located in the array to handle the additional capacity. Each substation is the terminus of nine cable routes, for a total of 18 cable routes in the array.

#### 3.2.1 Site conditions

The Gulf of Maine wind energy call area features water depths of approximately 100–300 m (Musial et al., 2023). We chose a constant water depth of 200 m for this reference array. The wind, wave, and current roses for the Gulf of Maine reference site conditions are shown in Fig. 15. The extreme load case conditions are described in Table 12.

#### 3.2.2 Mooring and cable design

Figure 16 shows the mooring and dynamic cable configuration for the Gulf of Maine.

The Gulf of Maine array design uses a three-line semitaut mooring system consisting of chain and polyester with drag-450 embedment anchors from Lozon et al. (2025). The chain section is approximately 500 m long, and the polyester section is 200 m long, with an anchoring radius of 700 m. Table 13 details the mooring configuration.

Figure 13. Final routing of intra-array cables into substation for the Humboldt Bay array in (a) plan view and (b) 3 dimensions

Table 12. Gulf of Maine extreme load case conditions

| Parameter            | DLC 1.6 | DLC 6.1 | SLC   |
|----------------------|---------|---------|-------|
| $H_S$ (m)            | 7.11    | 11.86   | 14.19 |
| $T_P$ (m)            | 12.2    | 15.75   | 17.23 |
| Current speed (m/s)  | 0.71    | 0.88    | 1.34  |
| Wind speed (m/s)     | 10.59   | 40.59   | 42.96 |
| Turbulence intensity | .06     | .05     | .05   |

The dynamic cable is a lazy-wave configuration, also adopted from Lozon et al. (2025). We directly implemented the 300 mm<sup>2</sup> cable conductor size from Lozon et al. (2025), and then we adapted the number of buoyancy modules for the larger conductors sizes. The 300 mm<sup>2</sup> cable includes 6 buoyancy modules over the buoyancy section, while the 630 mm<sup>2</sup> cable includes 10, and the 1000 mm<sup>2</sup> cable includes 14. All cables have a constant buoyancy section length of 60 m, meaning the buoyancy module spacing decreases as the conductor size increases. The design parameters for the three different cable conductor sizes are shown in Table 14.

**Figure 14.** Humboldt Bay optimized array layout: (a) wakes with a wind speed of 12 m/s from due north and (b) % wake losses with a wind speed of 12 m/s at each wind heading direction

**Table 13.** Gulf of Maine mooring line design adapted from Lozon et al. (2025)

| Parameter                 | Value                    |
|---------------------------|--------------------------|
| Anchoring radius (m)      | 700                      |
| Fairlead radius (m)       | 58                       |
| Fairlead depth (m)        | 14                       |
| Pretension (kN)           | 1205                     |
| Declination angle (°)     | 38.33                    |
| Line section 1 material   | 181.8 mm polyester       |
| Line section 1 length (m) | 199.8                    |
| Line section 2 material   | 155 mm R4 studless chain |
| Line section 2 length (m) | 497.7                    |

Figure 15. Gulf of Maine (a) wind, (b) wave, and (c) current roses (Biglu et al., 2024a)

Table 14. Gulf of Maine lazy-wave dynamic cable designs

| Parameter                                  |        | Value  |        |
|--------------------------------------------|--------|--------|--------|
| Conductor size (mm <sup>2</sup> )          | 300    | 630    | 1000   |
| Anchor point (m)                           | 205    | 205    | 205    |
| Total cable length (m)                     | 353.51 | 353.51 | 353.51 |
| Length of cable below buoyancy section (m) | 121.53 | 121.53 | 121.53 |
| Buoyancy section length (m)                | 60     | 60     | 60     |
| Midpoint of buoyancy section (m)           | 151.53 | 151.53 | 151.53 |
| Length of cable above buoyancy section (m) | 171.98 | 171.98 | 171.98 |
| Number of buoyancy modules                 | 6      | 10     | 14     |
| Buoyancy module spacing (m)                | 11.23  | 6.38   | 4.53   |
| Averaged diameter of buoyancy section (m)  | 0.30   | 0.40   | 0.46   |
| Averaged mass of buoyancy section (kg/m)   | 60.85  | 103.22 | 140.73 |

# 3.2.3 Optimized layout

We optimized the Gulf of Maine reference array to minimize the LCOE for 132 turbines within a 504.5 km<sup>2</sup> area. Table 15 shows the grid transformation design variables for the Gulf of Maine optimized reference array layout. The spacing in the x

Figure 16. Gulf of Maine mooring and dynamic cable system (Lozon et al., 2025)

direction is 1442 m, and the spacing in the y direction is 2564 m. The grid has a  $180^{\circ}$  rotation with a skew of  $-18^{\circ}$ , and all turbines are rotated to  $60^{\circ}$ .

The optimized reference array layout is shown in Fig. 17.

There are two substations in this array to accommodate the larger number of turbines. The substations are located at a slight offset from the center of the farm, with one closer to the northwest corner and one closer to the southeast corner. We chose these locations to reduce the lengths of cables connected in series.

Each substation mooring system features eight lines, with two on each corner spaced 20° apart. One side of each substation is free of cables to allow space for a maintenance vessel to approach. The intra-array cables enter at headings spaced 5° apart for each side. The substations are rotated 25°, which is 35° less than the turbine platforms. We chose this heading after the optimization process to prevent sharp angles when rerouting the cables entering the substation. Figure 18 shows a close-up view of this rerouting on the northeast substation.

Table 15. Gulf of Maine optimized reference array layout design variables

| Parameter                       | Value   |
|---------------------------------|---------|
| Grid $x$ spacing, $D'_x$ (m)    | 1442.0  |
| Grid $y$ spacing, $D'_y$ (m)    | 2563.7  |
| Grid $x$ translation, $x_0$ (m) | -1562.2 |
| Grid $y$ translation, $y_0$ (m) | 2359.9  |
| Grid rotation, $\alpha$ (°)     | 180.0   |
| Grid skew, $\beta$ (°)          | -18.3   |
| Platform rotation, $\gamma$ (°) | 60.0    |

Figure 17. Gulf of Maine array layout and cable routing in (a) plan view and (b) 3 dimensions

The algorithm described in Sect. 2.3.6 rerouted the dynamic cables to be at least 25° offset from the mooring line headings of their associated platforms to avoid clashing, and then it rerouted the static cables to follow the dynamic cable heading for an additional 300 m before routing toward the next platform to ensure that the mooring lines and cables would not cross.

Figure 19a visualizes the FLORIS wake model with winds at 12 m/s from the predominant wind direction, 205° clockwise from due north. Figure 19b shows a polar plot of the wake losses for a wind speed of 12 m/s at each angle with a 1° interval. Though some directions produce significant wake losses, the wake losses dramatically decrease with even slight changes in the wind direction. When comparing the major wake loss directions in subplot (b) to the uniform grid layout in subplot (a), the

Figure 18. Final routing of intra-array cables into substation for the Gulf of Maine array in (a) plan view and (b) 3 dimensions

largest wake losses are along the east-west directions at up to 35 % loss due to the small grid spacing in this direction. Figure 15a shows that the wind resource is limited in this direction, so there is limited impact on the AEP.

There are also notable wake losses in the northwest-southeast and northeast-southwest directions, which can be attributed to the cross-wise grid direction; however, these wake losses are less than 10 %. Near the predominant wind direction, there is a wake loss of less than 1 %. The layout largely avoids wake losses in directions with significant wind resource. The wind rose of the Gulf of Maine is more spread than that of Humboldt Bay, making it difficult to completely avoid wake losses in all relevant wind directions. Notably, the wind rose data used to calculate AEP used 5° direction intervals to balance computational efficiency with AEP accuracy. When those same 5° intervals are used to calculate wake loss percentages, no wake losses appear in the interval covering the predominant wind direction due to the drastic decay in the wake losses around a specific angle. This suggests that a more granular wind rose discretization might be needed to better capture the wake losses in the AEP calculations within layout optimizations.

The final values affecting the LCOE calculations in the optimization process are described in Table 16. These cost values reflect the final design, which includes the refined cable routing. The Gulf of Maine total anchor material costs are an order of magnitude less the total cable material costs. The cable material costs are 43 % less than the mooring system material costs. These costs are based on the cost curves in Sect. 2.3.3. The AEP, at nearly 10 TWh, is significantly larger than that of the Humboldt Bay design due to the increased number of turbines.

**Figure 19.** Gulf of Maine optimized array layout: (a) wakes with a wind speed of 12 m/s from the predominant wind direction and (b) wake losses at each direction with a wind speed of 12 m/s

Table 16. Gulf of Maine reference array layout AEP and mooring, cable, and anchor CapEx

| Performance Metric        | Value  |
|---------------------------|--------|
| AEP (GWh)                 | 9859.5 |
| Cable CapEx (\$M)         | 250.9  |
| Mooring lines CapEx (\$M) | 442.1  |
| Anchor CapEx (\$M)        | 15.7   |

# 495 3.3 Gulf of America

The Gulf of America array design features catenary mooring systems and lazy-wave dynamic cables. It has 67 turbines for a total capacity of 1.05 GW. The substation, located in the center of the array, is the terminus of nine cable routes.

Figure 20. Gulf of America (a) wind, (b) wave, and (c) current roses (Biglu et al., 2024a)

#### 3.3.1 Site conditions

The Gulf of America has a wide range of water depths within the federal exclusive economic zone. The wind energy call area developed for the Gulf of America is mostly shallow water suitable for fixed-bottom wind turbines, but some portions are deep enough (greater than 60 m) to require floating wind (Fuchs et al., 2023). We chose a water depth of 80 m for the Gulf of America reference array. The wind, wave, and current roses are shown in Fig. 20. The extreme load cases used to evaluate the mooring and cable designs are described in Table 17.

**Table 17.** Gulf of America extreme load case conditions

| Parameter            | DLC 1.6 | DLC 6.1 | SLC  |
|----------------------|---------|---------|------|
| $H_S$ (m)            | 5.5     | 6.8     | 7.4  |
| $T_P(m)$             | 10.8    | 11.9    | 12.5 |
| Current speed (m/s)  | 0.71    | 0.88    | 1.34 |
| Wind speed (m/s)     | 10.59   | 29.8    | 31.5 |
| Turbulence intensity | .06     | .05     | .05  |

Figure 21. Gulf of America mooring and cable configuration (Lozon et al., 2025)

#### 3.3.2 Mooring and cable design

We adopted the three-line catenary chain mooring system with drag-embedment anchors and lazy-wave dynamic cables developed in Lozon et al. (2025) for use in the Gulf of America reference array. The anchoring radius is 400 m with a total line length of 364.5 m. Figure 21 shows the mooring and dynamic cable configurations used in the Gulf of America reference array.

The dynamic cables used in the Gulf of America reference array are 80 m depth lazy-wave cable designs adopted from Lozon et al. (2025). From the original optimized design for the 300 mm<sup>2</sup> cable conductor size, we adapted the number of buoyancy modules for the larger sizes. The 300 mm<sup>2</sup> cable includes 5 buoyancy modules over the buoyancy section, while the 630 mm<sup>2</sup>

Table 18. Gulf of America mooring line design adapted from Lozon et al. (2025)

| Parameter             | Value                    |
|-----------------------|--------------------------|
| Anchoring radius (m)  | 400                      |
| Fairlead radius (m)   | 58                       |
| Fairlead depth (m)    | 14                       |
| Pretension (kN)       | 748                      |
| Declination angle (°) | 52.0                     |
| Line material         | 160 mm R4 studless chain |
| Line length (m)       | 364.5                    |

cable includes 8, and the 1000 mm<sup>2</sup> cable includes 12. All cables have a constant buoyancy section length of 50 m, meaning the buoyancy module spacing decreases as the conductor size increases. The cable design parameters are shown in Table 19.

Table 19. Gulf of America lazy-wave dynamic cable designs

| Parameter                                  |         | Value   |         |
|--------------------------------------------|---------|---------|---------|
| Conductor size (mm <sup>2</sup> )          | 300     | 630     | 1000    |
| Anchor point (m)                           | 125     | 125     | 125     |
| Total cable length (m)                     | 170.215 | 170.215 | 170.215 |
| Length of cable below buoyancy section (m) | 52.101  | 52.101  | 52.101  |
| Buoyancy section length (m)                | 50      | 50      | 50      |
| Midpoint of buoyancy section (m)           | 77.1    | 77.1    | 77.1    |
| Length of cable above buoyancy section (m) | 68.114  | 68.114  | 68.114  |
| Number of buoyancy modules                 | 5       | 8       | 12      |
| Buoyancy module spacing (m)                | 11.88   | 7.23    | 4.59    |
| Averaged diameter of buoyancy section (m)  | 0.290   | 0.386   | 0.463   |
| Averaged mass of buoyancy section (kg/m)   | 59.53   | 99.09   | 140.83  |

# 3.3.3 Optimized layout

We optimized the Gulf of America reference array layout to minimize the LCOE for 67 turbines in a 280.7 km<sup>2</sup> square lease area. The layout is a uniform grid with 1189 m spacing in the x direction and 3991 m spacing in the y direction. There is no grid rotation, but there is a 6° skew. Each turbine platform has a heading of 60.3°. Table 20 shows the grid transformation

Figure 22. Gulf of America array layout and cable routing in (a) plan view and (b) 3 dimensions

variables for the Gulf of America reference array layout. The southeast predominant wind direction led to a significantly larger spacing in the y direction than the x direction.

Table 20. Gulf of America optimized reference array layout design variables

| Parameter                | Value   |
|--------------------------|---------|
| Grid x spacing (m)       | 1188.9  |
| Grid $y$ spacing (m)     | 3991.2  |
| Grid $x$ translation (m) | -414.7  |
| Grid $y$ translation (m) | -3878.1 |
| Grid rotation (°)        | 0.0     |
| Grid skew (°)            | 6.0     |
| Platform rotation (°)    | 60.3    |

The optimized layout for the Gulf of America reference array is shown in Fig. 22.

It is notable that there are extra grid locations without turbines. The location of the unused grid points is based on the optimization process's method of filling in the grid, which removes turbines closest to the lease area boundary until the correct number of turbines are remaining. It is possible that the placement of these unused grid points and the substation in another

Figure 23. Final routing of intra-array cables into substation for the Gulf of America array in (a) plan view and (b) 3 dimensions

part of the grid could improve the AEP; however, these considerations are an additional variable that is out of the scope of this optimization. We placed the substation in the center of the array to reduce the cable lengths and sizes.

The substation mooring system features eight mooring lines, with two on each corner of the substation spaced 20° apart. Figure 23 provides a close-up view of the rerouting around the substation. Cables are routed to three sides, with three cables on each. In each side, cable headings entering the substation are spaced 5° apart to prevent clashing between cables. The substation is rotated 35.3°, 25° less than the turbine platforms. We chose this heading after the optimization process to prevent sharp angles when rerouting the cables entering the substation.

Figure 24a visualizes the Gulf of America FLORIS wake model with winds at 12 m/s from the southeast predominant wind direction. Figure 24b shows a polar plot of the wake losses, where the array wake losses were calculated for every angle at an interval of  $1^{\circ}$  when the wind speed is 12 m/s. When comparing the major wake loss directions in subplot (b) to the uniform grid layout in subplot (a), it is clear that the main wake loss directions are east-west along the x direction of the grid, as this direction affords the least distance between turbines. Comparing subplot (b) with the wind rose in Fig. 20a, the southeast predominant wind direction does not align with the major wake loss directions. Though there is significant wind coming from the south, which aligns with the y direction of the grid, there is no significant wake loss due to the large north-south spacing in the grid.

The final values affecting the LCOE calculations in the optimization process are described in Table 21. The cable material costs, at \$112M, are 31 % less than the total mooring material costs of \$163M. These costs reflect the material cost of the final

**Figure 24.** Gulf of America optimized array layout: (a) wakes with a wind speed of 12 m/s from the predominant wind direction and (b) % wake losses with a wind speed of 12 m/s at each wind heading direction

design, including the refined cable routing. The AEP is less than that of Humboldt Bay, which has the same total capacity, consistent with the reduced wind resource in the Gulf of America.

Table 21. Gulf of America reference array layout AEP and mooring, cable, and anchor CapEx

| Performance Metric        | Value  |
|---------------------------|--------|
| AEP (GWh)                 | 3681.9 |
| Cable CapEx (\$M)         | 112.5  |
| Mooring lines CapEx (\$M) | 163.1  |
| Anchor CapEx (\$M)        | 9.6    |

## 4 Conclusions

Floating wind reference array designs were developed for three representative regions of the United States while accounting for the site conditions of each area. Each design has a uniform grid array layout that is optimized to approximately minimize the LCOE. The designs include three-dimensional definitions of major components and systems – such as mooring lines, anchors,

dynamic cables, turbines, floating platforms, and substations – as well as the layout of these components and the routing of the static array cables.

The design approach combines the adaptation of the existing component designs and the optimization of the array layout along with additional fine-tuning steps. All designs use the common VolturnUS-S 15 MW reference floating wind turbine system and an existing floating substation design. Reference mooring lines, dynamic cables, and anchors were adopted from previous work. We adapted the dynamic cable designs for the larger conductor sizes needed by these arrays. We developed an array layout methodology that built upon previous work to improve efficiency, integrate substations in the uniform grid, and route to multiple substations in an array optimization. Spatial constraints were used to ensure the output array design was feasible. Intra-array cable routing was developed using three different conductor sizes for the unique layout of each array. Further routing adjustments were made with an algorithm developed to prevent cables from clashing with moorings and anchors, and manual adjustments were made to connect the intra-array dynamic cables to the substation at 5° intervals. These cable routing adjustment techniques create more realistic cable routing in the array layout designs. We confirmed the layout optimality of each array by checking the wake losses at each wind heading, and we found that the arrays largely avoid wake losses in the predominant wind directions.

The reference designs, and especially their optimized array layouts, provide examples of effective design characteristics for each region. The Humboldt Bay design uses taut mooring systems for cost efficiency in deep water. The large anchoring radius necessitates similar turbine spacings in each direction, despite the very directional wind resource. Wake losses are instead minimized by orienting the array layout at a diagonal to the predominant wind direction. The Gulf of Maine design uses semitaut mooring systems for cost efficiency in moderate water depths. Given the larger size of the proposed lease areas in the Gulf of Maine, we used a larger array capacity and two substations. The Gulf of Maine wind resource is relatively spread compared to Humboldt Bay, requiring avoidance of wake losses in multiple different directions. The wake losses are minimized with a larger y spacing and a significant grid skew angle. The Gulf of America design uses catenary mooring systems due to the shallow depth. The fairly directional wind resource and small anchoring radius allowed platforms to be tightly packed in the x direction, leaving large spacing in the y direction to avoid wake losses in the dominant wind direction.

The three reference designs are described in detail by publicly available design definition files, making the designs available for use in floating wind research and development projects where array-level scenarios are needed. These reference designs can serve as baselines for evaluating various floating wind innovations at the array scale or comparing with commercial-scale floating wind array designs. The designs can also be built upon or adapted, with portions of the design substituted to fit different locations, requirements, and research focuses.

The scope of the presented reference design methodology is limited to approximately optimizing uniform grid array layouts of selected floating turbine systems with a representative wind rose, lease area, water depth, and soil type for each region. Spatial constraints account for navigability and potential component clashing. Though this methodology was carefully chosen to yield practical reference designs for a variety of purposes, there are a few clear limitations to the presented approach. One limitation is the relatively coarse discretization of the wind rose directions, which was found to not capture all relevant wake losses due to their specific directionality. Another limitation is the superficial analysis of the optimization approach

performance; although the layouts perform well, it is likely that the optimizer did not find the true global minimum LCOE. Further, some highly site-specific factors relevant to commercial-scale floating wind array designs, such as grid interconnection details, were not in the scope of these reference designs.

Future work could evaluate the reference designs for many real-world factors that were not considered in detail for the presented methodology, such as varied bathymetry and sediment, export cable routing, installation, maintenance, and supply chain availability. The scope of the reference design approach could be expanded to consider these drivers for a more holistic design. For example, installation and maintenance techniques can be simulated and optimized for these layout designs, and constraints or estimations can be added to the design methodology to consider these factors in future array layout optimizations. Optimization approaches and settings can be compared and adjusted to achieve a more optimal or faster optimization.

Data availability. Complete reference array design descriptions are available at https://github.com/FloatingArrayDesign/ReferenceDesigns

### **Appendix A: Humboldt Bay Platform Positions**

The Humboldt Bay array layout platform positions are listed in Table A1. The mooring orientation is counterclockwise relative to a mooring line due north for turbines. For substations, the mooring orientation is relative to one pontoon facing each cardinal direction.

Table A1. Humboldt Bay array layout platform positions

| Platform   | X Position | Y Position | Orientation (°) |
|------------|------------|------------|-----------------|
| Turbine 1  | 9051       | 9170       | 3               |
| Turbine 2  | 8278       | 6809       | 63              |
| Turbine 3  | 6861       | 9323       | 63              |
| Turbine 4  | 9759       | 7913       | 63              |
| Turbine 5  | 6088       | 6962       | 3               |
| Turbine 6  | 6797       | 5705       | 63              |
| Turbine 7  | 8342       | 10427      | 63              |
| Turbine 8  | 8986       | 5552       | 3               |
| Turbine 9  | 10467      | 6656       | 3               |
| Turbine 10 | 10532      | 10274      | 3               |
| Turbine 11 | 6153       | 10580      | 3               |
| Turbine 12 | 5380       | 8219       | 63              |
| Turbine 13 | 11176      | 5399       | 63              |
| Turbine 14 | 11240      | 9017       | 63              |
| Turbine 15 | 4672       | 9476       | 3               |
| Turbine 16 | 4607       | 5858       | 3               |
| Turbine 17 | 5316       | 4601       | 63              |
| Turbine 18 | 9823       | 11531      | 63              |
| Turbine 19 | 7505       | 4448       | 3               |
| Turbine 20 | 7634       | 11684      | 3               |
| Turbine 21 | 9695       | 4295       | 63              |
| Turbine 22 | 5444       | 11837      | 63              |
| Turbine 23 | 11884      | 4142       | 3               |
| Turbine 24 | 11949      | 7760       | 3               |
| Turbine 25 | 12013      | 11378      | 3               |
| Turbine 26 | 3963       | 10733      | 63              |
| Turbine 27 | 3899       | 7115       | 63              |
| Turbine 28 | 3835       | 3497       | 63              |
| Turbine 29 | 11304      | 12635      | 63              |
| Turbine 30 | 6024       | 3344       | 3               |
| Turbine 31 | 12657      | 6503       | 63              |
| Turbine 32 | 12721      | 10121      | 63              |
| Turbine 33 | 3255       | 11990      | 3               |
|            |            |            |                 |

Table A1 – continued from previous page

| Table A1 – continued from previous page |            |            |                 |
|-----------------------------------------|------------|------------|-----------------|
| Platform                                | X Position | Y Position | Orientation (°) |
| Turbine 34                              | 9115       | 12788      | 3               |
| Turbine 35                              | 8214       | 3191       | 63              |
| Turbine 36                              | 3190       | 8372       | 3               |
| Turbine 37                              | 3126       | 4754       | 3               |
| Turbine 38                              | 6925       | 12941      | 63              |
| Turbine 39                              | 10403      | 3038       | 3               |
| Turbine 40                              | 4736       | 13094      | 3               |
| Turbine 41                              | 12593      | 2885       | 63              |
| Turbine 42                              | 13365      | 5246       | 3               |
| Turbine 43                              | 13430      | 8864       | 3               |
| Turbine 44                              | 2546       | 13247      | 63              |
| Turbine 45                              | 13494      | 12482      | 3               |
| Turbine 46                              | 2482       | 9629       | 63              |
| Turbine 47                              | 2418       | 6011       | 63              |
| Turbine 48                              | 2354       | 2393       | 63              |
| Turbine 49                              | 12785      | 13739      | 63              |
| Turbine 50                              | 4543       | 2240       | 3               |
| Turbine 51                              | 10596      | 13892      | 3               |
| Turbine 52                              | 6733       | 2087       | 63              |
| Turbine 53                              | 8406       | 14045      | 63              |
| Turbine 54                              | 8922       | 1934       | 3               |
| Turbine 55                              | 14074      | 3989       | 63              |
| Turbine 56                              | 14138      | 7606       | 63              |
| Turbine 57                              | 6217       | 14198      | 3               |
| Turbine 58                              | 14202      | 11224      | 63              |
| Turbine 59                              | 11112      | 1781       | 63              |
| Turbine 60                              | 1774       | 10886      | 3               |
| Turbine 61                              | 1709       | 7268       | 3               |
| Turbine 62                              | 4027       | 14351      | 63              |
| Turbine 63                              | 1645       | 3651       | 3               |
| Turbine 64                              | 13301      | 1628       | 3               |
| Turbine 65                              | 1838       | 14504      | 3               |
| Turbine 66                              | 14782      | 2731       | 3               |
| Turbine 67                              | 14267      | 14842      | 63              |
|                                         |            |            |                 |

Table A1 – continued from previous page

| Platform   | X Position | Y Position | Orientation (°) |
|------------|------------|------------|-----------------|
| Substation | 7569       | 8066       | 3               |

## **Appendix B: Gulf of Maine Platform Positions**

The Gulf of Maine array layout platform positions are listed in Table B1. The mooring orientation is counterclockwise relative to a mooring line due north for turbines. For substations, the mooring orientation is relative to one pontoon facing each cardinal direction.

Table B1. Gulf of Maine array layout platform positions

| Platform   | X Position | Y Position | Orientation (°) |
|------------|------------|------------|-----------------|
| Turbine 1  | 21549      | 21282      | 60              |
| Turbine 2  | 20107      | 21282      | 60              |
| Turbine 3  | 18665      | 21282      | 60              |
| Turbine 4  | 17223      | 21282      | 60              |
| Turbine 5  | 15781      | 21282      | 60              |
| Turbine 6  | 14339      | 21282      | 60              |
| Turbine 7  | 12897      | 21282      | 60              |
| Turbine 8  | 11455      | 21282      | 60              |
| Turbine 9  | 10013      | 21282      | 60              |
| Turbine 10 | 8571       | 21282      | 60              |
| Turbine 11 | 7129       | 21282      | 60              |
| Turbine 12 | 5687       | 21282      | 60              |
| Turbine 13 | 4245       | 21282      | 60              |
| Turbine 14 | 2803       | 21282      | 60              |
| Turbine 15 | 1361       | 21282      | 60              |
| Turbine 16 | 20953      | 18718      | 60              |
| Turbine 17 | 19511      | 18718      | 60              |
| Turbine 18 | 18069      | 18718      | 60              |
| Turbine 19 | 16627      | 18718      | 60              |
| Turbine 20 | 15185      | 18718      | 60              |
| Turbine 21 | 13743      | 18718      | 60              |
| Turbine 22 | 12301      | 18718      | 60              |
| Turbine 23 | 10859      | 18718      | 60              |
| Turbine 24 | 9418       | 18718      | 60              |
| Turbine 25 | 7976       | 18718      | 60              |
| Turbine 26 | 6534       | 18718      | 60              |
| Turbine 27 | 5092       | 18718      | 60              |
| Turbine 28 | 3650       | 18718      | 60              |
| Turbine 29 | 2208       | 18718      | 60              |
| Turbine 30 | 766        | 18718      | 60              |
| Turbine 31 | 21800      | 16154      | 60              |
| Turbine 32 | 20358      | 16154      | 60              |
| Turbine 33 | 18916      | 16154      | 60              |
|            |            |            |                 |

Table B1 – continued from previous page

|                  |            | TAR MA     |                 |
|------------------|------------|------------|-----------------|
| Platform<br>———— | X Position | Y Position | Orientation (°) |
| Turbine 34       | 17474      | 16154      | 60              |
| Turbine 35       | 16032      | 16154      | 60              |
| Turbine 36       | 14590      | 16154      | 60              |
| Turbine 37       | 13148      | 16154      | 60              |
| Turbine 38       | 11706      | 16154      | 60              |
| Turbine 39       | 10264      | 16154      | 60              |
| Turbine 40       | 8822       | 16154      | 60              |
| Turbine 41       | 7380       | 16154      | 60              |
| Turbine 42       | 5938       | 16154      | 60              |
| Turbine 43       | 4496       | 16154      | 60              |
| Turbine 44       | 3054       | 16154      | 60              |
| Turbine 45       | 1612       | 16154      | 60              |
| Turbine 46       | 21204      | 13590      | 60              |
| Turbine 47       | 19762      | 13590      | 60              |
| Turbine 48       | 18320      | 13590      | 60              |
| Turbine 49       | 16878      | 13590      | 60              |
| Turbine 50       | 15436      | 13590      | 60              |
| Turbine 51       | 13994      | 13590      | 60              |
| Turbine 52       | 12552      | 13590      | 60              |
| Turbine 53       | 11110      | 13590      | 60              |
| Turbine 54       | 9668       | 13590      | 60              |
| Turbine 55       | 6784       | 13590      | 60              |
| Turbine 56       | 5342       | 13590      | 60              |
| Turbine 57       | 3900       | 13590      | 60              |
| Turbine 58       | 2458       | 13590      | 60              |
| Turbine 59       | 1016       | 13590      | 60              |
| Turbine 60       | 20609      | 11027      | 60              |
| Turbine 61       | 19167      | 11027      | 60              |
| Turbine 62       | 17725      | 11027      | 60              |
| Turbine 63       | 16283      | 11027      | 60              |
| Turbine 64       | 14841      | 11027      | 60              |
| Turbine 65       | 13399      | 11027      | 60              |
| Turbine 66       | 11957      | 11027      | 60              |
| Turbine 67       | 10515      | 11027      | 60              |
|                  |            |            |                 |

Table B1 – continued from previous page

| Table       | Table B1 – continued from previous page |            |                 |  |
|-------------|-----------------------------------------|------------|-----------------|--|
| Platform    | X Position                              | Y Position | Orientation (°) |  |
| Turbine 68  | 9073                                    | 11027      | 60              |  |
| Turbine 69  | 7631                                    | 11027      | 60              |  |
| Turbine 70  | 6189                                    | 11027      | 60              |  |
| Turbine 71  | 4747                                    | 11027      | 60              |  |
| Turbine 72  | 3305                                    | 11027      | 60              |  |
| Turbine 73  | 1863                                    | 11027      | 60              |  |
| Turbine 74  | 21455                                   | 8463       | 60              |  |
| Turbine 75  | 20013                                   | 8463       | 60              |  |
| Turbine 76  | 18571                                   | 8463       | 60              |  |
| Turbine 77  | 17129                                   | 8463       | 60              |  |
| Turbine 78  | 15687                                   | 8463       | 60              |  |
| Turbine 79  | 12803                                   | 8463       | 60              |  |
| Turbine 80  | 11361                                   | 8463       | 60              |  |
| Turbine 81  | 9919                                    | 8463       | 60              |  |
| Turbine 82  | 8477                                    | 8463       | 60              |  |
| Turbine 83  | 7035                                    | 8463       | 60              |  |
| Turbine 84  | 5593                                    | 8463       | 60              |  |
| Turbine 85  | 4151                                    | 8463       | 60              |  |
| Turbine 86  | 2709                                    | 8463       | 60              |  |
| Turbine 87  | 1267                                    | 8463       | 60              |  |
| Turbine 88  | 20859                                   | 5899       | 60              |  |
| Turbine 89  | 19417                                   | 5899       | 60              |  |
| Turbine 90  | 17975                                   | 5899       | 60              |  |
| Turbine 91  | 16533                                   | 5899       | 60              |  |
| Turbine 92  | 15091                                   | 5899       | 60              |  |
| Turbine 93  | 13650                                   | 5899       | 60              |  |
| Turbine 94  | 12208                                   | 5899       | 60              |  |
| Turbine 95  | 10766                                   | 5899       | 60              |  |
| Turbine 96  | 9324                                    | 5899       | 60              |  |
| Turbine 97  | 7882                                    | 5899       | 60              |  |
| Turbine 98  | 6440                                    | 5899       | 60              |  |
| Turbine 99  | 4998                                    | 5899       | 60              |  |
| Turbine 100 | 3556                                    | 5899       | 60              |  |
| Turbine 101 | 2114                                    | 5899       | 60              |  |
|             |                                         |            |                 |  |

Table B1 - continued from previous page

| 1able        | Table B1 – continued from previous page |            |                 |  |
|--------------|-----------------------------------------|------------|-----------------|--|
| Platform     | X Position                              | Y Position | Orientation (°) |  |
| Turbine 102  | 672                                     | 5899       | 60              |  |
| Turbine 103  | 21706                                   | 3335       | 60              |  |
| Turbine 104  | 20264                                   | 3335       | 60              |  |
| Turbine 105  | 18822                                   | 3335       | 60              |  |
| Turbine 106  | 17380                                   | 3335       | 60              |  |
| Turbine 107  | 15938                                   | 3335       | 60              |  |
| Turbine 108  | 14496                                   | 3335       | 60              |  |
| Turbine 109  | 13054                                   | 3335       | 60              |  |
| Turbine 110  | 11612                                   | 3335       | 60              |  |
| Turbine 111  | 10170                                   | 3335       | 60              |  |
| Turbine 112  | 8728                                    | 3335       | 60              |  |
| Turbine 113  | 7286                                    | 3335       | 60              |  |
| Turbine 114  | 5844                                    | 3335       | 60              |  |
| Turbine 115  | 4402                                    | 3335       | 60              |  |
| Turbine 116  | 2960                                    | 3335       | 60              |  |
| Turbine 117  | 1518                                    | 3335       | 60              |  |
| Turbine 118  | 21110                                   | 772        | 60              |  |
| Turbine 119  | 19668                                   | 772        | 60              |  |
| Turbine 120  | 18226                                   | 772        | 60              |  |
| Turbine 121  | 16784                                   | 772        | 60              |  |
| Turbine 122  | 15342                                   | 772        | 60              |  |
| Turbine 123  | 13900                                   | 772        | 60              |  |
| Turbine 124  | 12458                                   | 772        | 60              |  |
| Turbine 125  | 11016                                   | 772        | 60              |  |
| Turbine 126  | 9574                                    | 772        | 60              |  |
| Turbine 127  | 8132                                    | 772        | 60              |  |
| Turbine 128  | 6690                                    | 772        | 60              |  |
| Turbine 129  | 5248                                    | 772        | 60              |  |
| Turbine 130  | 3806                                    | 772        | 60              |  |
| Turbine 131  | 2364                                    | 772        | 60              |  |
| Turbine 132  | 923                                     | 772        | 60              |  |
| Substation 1 | 8226                                    | 13590      | 25              |  |
| Substation 2 | 14245                                   | 8463       | 25              |  |
|              |                                         |            |                 |  |

# **Appendix C: Gulf of America Platform Positions**

The Gulf of America array layout turbine positions are listed in Table C1. The mooring orientation is counterclockwise relative to a mooring line due north.

Table C1. Gulf of America array layout platform positions

| —————————————————————————————————————— | X Position | Y Position | Orientation (°) |
|----------------------------------------|------------|------------|-----------------|
|                                        |            |            |                 |
| Turbine 1                              | 411        | 508        | 60              |
| Turbine 2                              | 1599       | 508        | 60              |
| Turbine 3                              | 2788       | 508        | 60              |
| Turbine 4                              | 3977       | 508        | 60              |
| Turbine 5                              | 5166       | 508        | 60              |
| Turbine 6                              | 6355       | 508        | 60              |
| Turbine 7                              | 7544       | 508        | 60              |
| Turbine 8                              | 8733       | 508        | 60              |
| Turbine 9                              | 9922       | 508        | 60              |
| Turbine 10                             | 11111      | 508        | 60              |
| Turbine 11                             | 12300      | 508        | 60              |
| Turbine 12                             | 13488      | 508        | 60              |
| Turbine 13                             | 14677      | 508        | 60              |
| Turbine 14                             | 15866      | 508        | 60              |
| Turbine 15                             | 830        | 4500       | 60              |
| Turbine 16                             | 2018       | 4500       | 60              |
| Turbine 17                             | 3207       | 4500       | 60              |
| Turbine 18                             | 4396       | 4500       | 60              |
| Turbine 19                             | 5585       | 4500       | 60              |
| Turbine 20                             | 6774       | 4500       | 60              |
| Turbine 21                             | 7963       | 4500       | 60              |
| Turbine 22                             | 9152       | 4500       | 60              |
| Turbine 23                             | 10341      | 4500       | 60              |
| Turbine 24                             | 11530      | 4500       | 60              |
| Turbine 25                             | 12719      | 4500       | 60              |
| Turbine 26                             | 13907      | 4500       | 60              |
| Turbine 27                             | 15096      | 4500       | 60              |
| Turbine 28                             | 16285      | 4500       | 60              |
| Turbine 29                             | 1249       | 8491       | 60              |
| Turbine 30                             | 2437       | 8491       | 60              |
| Turbine 31                             | 3626       | 8491       | 60              |
| Turbine 32                             | 4815       | 8491       | 60              |
| Turbine 33                             | 6004       | 8491       | 60              |
|                                        |            |            |                 |

Table C1 – continued from previous page

| Platform   | X Position | Y Position | Orientation (°) |
|------------|------------|------------|-----------------|
| Turbine 34 | 7193       | 8491       | 60              |
| Turbine 35 | 9571       | 8491       | 60              |
| Turbine 36 | 10760      | 8491       | 60              |
| Turbine 37 | 11949      | 8491       | 60              |
| Turbine 38 | 13138      | 8491       | 60              |
| Turbine 39 | 14326      | 8491       | 60              |
| Turbine 40 | 15515      | 8491       | 60              |
| Turbine 41 | 479        | 12482      | 60              |
| Turbine 42 | 1668       | 12482      | 60              |
| Turbine 43 | 2857       | 12482      | 60              |
| Turbine 44 | 4045       | 12482      | 60              |
| Turbine 45 | 5234       | 12482      | 60              |
| Turbine 46 | 6423       | 12482      | 60              |
| Turbine 47 | 7612       | 12482      | 60              |
| Turbine 48 | 8801       | 12482      | 60              |
| Turbine 49 | 9990       | 12482      | 60              |
| Turbine 50 | 11179      | 12482      | 60              |
| Turbine 51 | 12368      | 12482      | 60              |
| Turbine 52 | 13557      | 12482      | 60              |
| Turbine 53 | 14745      | 12482      | 60              |
| Turbine 54 | 15934      | 12482      | 60              |
| Turbine 55 | 2087       | 16473      | 60              |
| Turbine 56 | 3276       | 16473      | 60              |
| Turbine 57 | 4464       | 16473      | 60              |
| Turbine 58 | 5653       | 16473      | 60              |
| Turbine 59 | 6842       | 16473      | 60              |
| Turbine 60 | 8031       | 16473      | 60              |
| Turbine 61 | 9220       | 16473      | 60              |
| Turbine 62 | 10409      | 16473      | 60              |
| Turbine 63 | 11598      | 16473      | 60              |
| Turbine 64 | 12787      | 16473      | 60              |
| Turbine 65 | 13976      | 16473      | 60              |
| Turbine 66 | 15164      | 16473      | 60              |
| Turbine 67 | 16353      | 16473      | 60              |

Table C1 – continued from previous page

| Platform     | X Position | Y Position | Orientation (°) |
|--------------|------------|------------|-----------------|
| Substation 1 | 8382       | 8491       | 35              |

Acknowledgements. This work was authored by NREL for the U.S. Department of Energy (DOE) under Contract No. DE-AC36-08GO28308. Funding provided by the U.S. Department of Energy Office of Energy Efficiency and Renewable Energy Wind Energy Technologies Office. The views expressed in the article do not necessarily represent the views of the DOE or the U.S. Government. The U.S. Government retains and the publisher, by accepting the article for publication, acknowledges that the U.S. Government retains a nonexclusive, paid-up, irrevocable, worldwide license to publish or reproduce the published form of this work, or allow others to do so, for U.S. Government purposes.

A portion of this research was performed using computational resources sponsored by the U.S. Department of Energy's Office of Energy Efficiency and Renewable Energy and located at NREL.

Author contributions. Leah Sirkis, Ericka Lozon, and Matthew Hall developed the methodology and wrote, edited, and reviewed the manuscript. Leah Sirkis and Ericka Lozon performed case studies, analysis, data curation, and visualization. Matthew Hall contributed supervision, project administration, funding acquisition, and conceptualization.

*Competing interests.* The authors declare that they have no known competing financial interests or personal relationships that could have appeared to influence the work reported in this paper.

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
