# Peer review of "Reference Floating Wind Array Designs for Three Representative Regions"

_Wind Energy Science, 2025_

## Author Comment (AC3)

The authors thank the referees for their careful review and insightful comments on this manuscript. We have addressed these comments and questions and included several additions to the manuscript, which are described below. Reviewer comments are shown in italicized *blue*, while our response is in black text.

**Reviewer 1**

1) *This manuscript shows the results of a layout optimization exercise for three floating wind turbine sites. The mooring system and initial power cable designs were developed in previous work, such that the present work only addresses the layout optimization. The assumptions are clear and the results are believable given the assumptions and approach, but the scientific contribution of the work is not very clear to me. What research questions does this work address? The novelty of the procedure only seems to be the need for buffer zones which would not generally be considered for a bottom-fixed layout. The procedure for cable routing is also already described in previous work, so it is not very clear to me what is new here.*

> Response:
>
> The main scientific contribution of the work is to provide detailed open-source reference floating array designs for three representative regions. Reference designs provide an important starting point or basis of comparison for follow-on research. Currently, no floating wind farm reference designs exist. The methodology in this paper also expands on previous work with several novel contributions.
>
> 1. We apply detailed dynamic and static cable routing to avoid obstacles such as moorings, anchors, and other cables. The cable routing process is primarily algorithmic, with some manual cable re-routings into the substations.
> 2. Substations are included within the uniform array during the optimization process, allowing the general cable routing to be performed at each iteration and included in the levelized cost of energy calculations.
> 3. Dynamic cable designs from Lozon et al. were adapted for larger conductor sizes of 630 and 1000 mm$^2$, in order to meet the power transmission needs of GW-scale wind farms. These new designs were tested with design load case 1.6, 6.1 and survival load case dynamic simulations in OpenFAST to ensure tensions and curvature meet standard requirements.
> 4. The optimization methodology from previous work was expanded to handle placement of and cable routing to multiple substations, such as in the presented Gulf of Maine array, with constraints to ensure the substations are not loaded over their maximum capacity.
>
> We have updated Figure 1 and added clarifying text throughout the document to more clearly portray the novel work in this manuscript. These changes include:
>
> - Added a step in the array design process flowchart in Figure 1 to show the intra-array cable design process, and expanded the cable routing adjustment step to include substeps that more accurately portray the work performed for this paper.

- Added text in the beginning of Sect. 2 describing the selection of cable conductor sizes from an initial layout and cable routing, and adaptation of cable designs from Lozon et al. for selected conductor sizes
- Added text in beginning of Sect. 2 to clarify that the cable routing adjustment is an algorithmic process with some manual alterations leading in to the substation.
- Added text in Sect. 2.2.5 to clarify that OpenFAST simulations were performed of the full FOWT system including each power cable design for extreme load cases

Text was added to the conclusion to include the power cable design and layout in the stated scope of the reference design methodology

The manuscript type has been changed to a Data Description article instead of a Research article, which better reflects the key contribution of open-source reference design descriptions.

2) *The premise of the optimization is that a uniform grid layout shall be obtained for a uniform water depth for each location. The procedure assumes that the mooring and cable design are already selected, while only layout optimization is performed here. It is unclear to me how this division can be included in a practical design situation, as the water depth in general will vary throughout the farm and require modifications to the mooring systems.*

Response:

The authors chose a uniform depth for each layout in this case because the lease areas are designed to be generally representative of a region rather than use the specific bathymetry and soil of one location in the region. By avoiding site-specific bathymetry and soil, the authors intended for the design to be applied and altered as necessary by potential future users for specific locations within the region, as stated in Sect. 2, 2.1, and the final paragraph of Sect. 4. The authors have added links to the bathymetry and soil dataset of each region on the GitHub ReadMe page to allow future researchers to adapt the designs for the bathymetry of a specific site within the region.

For realistic sites with variable bathymetry, it is common to have relatively standardized mooring and cable designs throughout the farm to simplify manufacturing, installation, and design processes. We have added the following text to the conclusion to describe these options and clarify the mooring and cable designs are standardized baselines that can be altered for specific sites:

"The designs can be applied and altered as necessary for a specific location within the region; for example, the mooring and cable designs used in the present work can be used as a standardized baseline design to be altered for site-specific bathymetry. Mooring designs can be adapted for small changes in water depth by increasing section lengths while maintaining horizontal pretension, and dynamic cable designs can be adapted for bathymetry by increasing cable length and adjusting buoyancy sections to maintain cable profile."

Additionally, the methodology adapts previously developed cable designs for larger conductor sizes required to meet power transmission needs throughout the array. This process includes performing critical load case simulations in OpenFAST to check design compliance with standards. This provides an example of the methodology for adapting a single dynamic cable design for the needs of a GW-scale farm. To better reflect this, we have updated Figure 1 and text in Sect. 2 in the manuscript to provide clarity on the work performed in this paper, including the cable design process.

3) *Furthermore, in Fig. 1, it seems that the cable sizes are input to step 4 separately from the cable design in step 3 – aren't these taken from step 3?*

Response:

The necessary cable sizes were determined from an initial layout and cable routing for a 1 GW farm. Step 3 (previous work) developed the 300 mm$^2$ dynamic cable designs for each region. A new step has been added in Figure 1 after step 3 to describe the process (performed in the present work) of selecting larger cable conductor sizes required for power transmission needs in the array, and then adapting the 300 mm$^2$ dynamic cable design from step 3 for the selected larger conductor sizes. The selected cable conductor sizes were then input to the optimization. The dynamic cable 3D designs were implemented after the optimization in the cable routing adjustment step, as the optimization used 2D cable lengths only. The cable sizes were kept constant for all three regions for consistency. Though the Gulf of Maine design capacity is 1.98 GW, there are two substations used to keep the number of turbines per substation approximately constant.

We have updated Figure 1 to improve clarity in the process by adding a new step 4, describing the cable size selection and dynamic cable design adaptation for larger conductor sizes. We also clarify in step 3 the conductor size of the dynamic cable design developed in previous work to differentiate between the previous and current cable design work. Steps 5 and 6 for layout optimization and cable routing adjustment were also expanded to clarify the substeps within those processes.

4) *There are also some notable shortcomings in the calculation of both the objective and constraints in the layout optimization. For the objective function, the AEP calculation is very sensitive to the number of wind directions considered, as the authors recognize. It does not appear that the sensitivity to this selection has been assessed for either the optimization process or the final designs.*

Response:

We have added sensitivity studies on the number of wind directions for each of the final designs in Sections 3.1.3, 3.2.3, and 3.3.3. We added plots comparing AEP values for various wind direction discretizations for each final layout (Figures 15, 21, and 27). The results of the sensitivity studies justify the selection of the 5˚ interval used in the layout optimizations. This is discussed in added text in Sections 3.1.3, 3.2.3, 3.3.3, and 4.

5) *For the fatigue and extreme response criteria, the frequency-domain tool RAFT has been applied. Has the applicability of a linearized tool for 500-year response conditions been assessed? The conditions for linearization are not likely to be met in extreme conditions.*

Response:

The authors would like to clarify that the frequency domain tool RAFT was not used to perform loads analysis for the floating wind turbine and its mooring systems and cables. In previous work, OpenFAST was used for fatigue analysis of the mooring system and extreme loads analysis of both the mooring design and the 300 mm$^2$ dynamic cable design, as discussed in Sect. 2.2.3 and 2.2.5. In the present work, OpenFAST was used for extreme loads analysis of the 630 and 1000 mm$^2$ dynamic cable designs, as mentioned in Sect. 2.2.5. The frequency domain model RAFT was used only to evaluate the floating substation design. It was evaluated in extreme 500-year wind, wave, and current to check the adequacy of the mooring system. The floating substation platform was adapted from other work and its dynamic performance was not the focus of the present work.

We have added the following sentence to Sect. 2.2.2 regarding the floating substation to clarify that dynamic performance of the substation platform is not a focus of the work:

"The dynamic performance of the substation platform, which was designed and evaluated in Jorge Alcantara (2023), is not a focus of the present work"

6) *Furthermore, the choice of extreme weather conditions, particularly for the GoM/GoA site which is expected to be dominated by hurricanes, deserves a bit more attention. (I have not thoroughly reviewed the reference which provides the background for these choices, but the Hs, Tp, and Uw for the SLC seem a bit low compared to what I might expect in a hurricane-prone area).*

Response:

We have added text to the GoM/GoA site conditions section of the article (Sect. 3.3.1) to clarify that the dataset used consistent methodologies across all regions for determining extreme values, which has limitations for capturing tropical cyclone conditions. The added text is as follows:

"The extreme load case parameters in this reference site dataset use the same approach as the other sites in Biglu et al. (2024b). This approach, which involves fitting probability distributions to the maxima or peaks in time series data, is a simplification that is not well suited for the extreme tropical cyclone conditions that can occur in this region. Designing specifically for tropical cyclone conditions was left for future work because the intent of the reference array designs is to suit the already defined reference site conditions."

7) *The optimization algorithm is described in limited detail. A PSO approach was used, but the number of particles, constraint handling, and convergence criteria are not described.*

Response:

We have included more information in the manuscript on the optimization algorithm and constraint handling in response to this comment. Text was added to Sect. 2.3.5 to clarify the number of particles and iterations. The following text was included:

"For this work, we used a swarm size of 200 evaluated for a minimum of 100 iterations. The Gulf of Maine and Gulf of America optimizations were run for 100 iterations, while the Humboldt Bay optimization was run for 364 iterations due to the increased complexity of the layout which increased the required number of iterations to achieve a general convergence. The evolution of the particle positions and best solution were monitored throughout the optimizations. In the case of the Gulf of Maine and Gulf of America optimizations, the optimization was unable to find better solutions well before 100 iterations and the swarm's particle positions were concentrated around the swarm's best known solution, so only 100 iterations were used for those designs. In the case of the Humboldt Bay optimization, new best solutions were frequently being discovered by the optimization at around the 100 iteration mark, so the optimization run time and number of iterations was increased until new iterations were consistently unable to determine a better solution."

The goal of this work was to develop and present detailed, open-source array layout designs that approximately minimize the LCOE while meeting the constraints and design requirements of each region; therefore, a detailed study of the optimization algorithms, settings, and convergence criteria that would lead to the global minimum LCOE was considered out of scope.

The constraints, described in Sect. 2.3.4, are checked for each layout considered, and a failed constraint prevents that solution from affecting the particle. The following text was added to the manuscript in Sect. 2.3.5 to provide clarity:

"If a particle's position fails any constraints, the particle's best position will not be updated and the particle is not considered when updating the overall swarm's best position at the end of the iteration. Therefore, the optimizer does not allow the solutions of non-feasible positions to influence future movement of the particle or swarm."

8) *The writing is generally pretty clear, though it could be a bit less colloquial in some instances ("pulled directly from…") and tense can vary in some parts of the manuscript. There can also be some confusion between what is meant by "we chose" vs. "the optimizer chose".*

Response:

The authors have added clarity through the document to differentiate between decisions made by the author and decisions made by the optimizer. For example, text was added to Sect. 3.1.3, 3.2.3, and 3.3.3 to clarify that the substation placement decision for each array was chosen by the authors to be central to the array, while the optimizer chose the specific substation location to be within the uniform grid at the grid point closest to the center of the array. The manuscript was reviewed to ensure accuracy and consistency of tense. Past tense is used for previous work or a previous step in the process being described, while present tense is used for current work to aid in differentiation.

**Reviewer 2**

1) *I see that the paper is registered as a "research article". While this has no great drawbacks, I invite you to consider switching it to a "Data description" paper since I think this fits better the nature of the study and would put it in a better spotlight. Maybe this can be a topic of discussion with the Editors.*

   Response:

   The manuscript type has been changed to a Data Description article based on this comment, as the authors agree this article type better represents the work.

2) *Is there any additional info on the real bathymetry in the site and the type of seabed available? You clearly stated that these are not accounted for at this stage, but adding such info (e.g., in the GitHub repository) could be useful to other researchers willing to further optimize the layouts in the future*

   Response:

   We have made a variety of edits to the GitHub page and manuscript to address this comment:

   - A link to the dataset containing bathymetry and seabed sediment data for each region has been added to the GitHub ReadMe page.
   - A sentence was added to the manuscript text in Sect. 2.1 to clarify that the bathymetry and soil data is in the dataset Biglu et al. (2024b):

     "The representative depth and soil type are based on the bathymetry and soil data found in Biglu et al. 2024b"

   - Sentences have been added to the conclusion describing future work of adjustment to mooring and cable design to meet site-specific bathymetry of a location within the region:
     "The designs can be applied and altered as necessary for a specific location within the region; for example, the mooring and cable designs used in the present work can be used as a standardized baseline design to be altered for site-specific bathymetry."

3) *Another piece of information that could be very useful to maximize the future exploitation of this study is represented by more detailed metocean conditions, including waves and wind-wave misalignment. The selected metocean conditions would deserve more attention also in the paper, as they are not addressed in a fully convincing way. If not fully measured, you could reconstruct them using for example the approach proposed in DOI: 10.1088/1742-6596/2385/1/012117*

   Response:

   We have made various edits to give greater clarity in response to this comment. These include:

- Clarifying in Sect. 2.1 and 2.2.3 that more detailed metocean condition information is available in the cited reference site condition dataset,
- Clarifying in Sect. 2.2.3 that the mooring system designs account for ultimate and fatigue metocean characteristics (including misalignment), and in Sect. 2.1 that the new design work presented in this paper focuses on AEP and select ultimate load cases.
- Adding text to Sect. 2.2.3 to state that 0° wind-wave misalignment was used for extreme load cases in order to obtain peak loading, while wind-wave misalignment was considered in fatigue load cases.
- The metocean datasets are now linked in the GitHub ReadMe file, where fatigue load cases considered are listed in the Summary spreadsheet of each dataset.

By relying on previously developed metocean datasets, we keep the scope contained but give readers access to more detailed metocean data that could be used for further exploration or design optimization while following the same site definition.

We have expanded Section 2.1 to give more information about how the metocean data are used and to note that these reference site conditions facilitate deeper future analysis:

"As detailed in Sect. 2.2.3, the mooring systems were designed for both ultimate and fatigue loads, relying on the processed extreme and fatigue metocean data from Biglu et al. The dynamic power cables were designed for extreme loads.

In the present work, we developed wind roses for each site from the same site condition dataset (Biglu et al. 2024b) for use during the array layout optimization process. We also used the extreme metocean data in load cases to check dynamic power cable designs and the floating substation designs. By adhering to the existing well-defined reference site conditions, additional research on the reference designs can look more deeply into various load cases of interest."

4) *Selected cost should be put in context, ideally providing more references and realistic "ranges" of variation of such costs for the time being*

Response:

We have added a table of cost comparisons between the component cost coefficients used in this manuscript and values found in literature (Table 6) to Sect. 2.3.3. We have also added discussion with the table to contextualize the cost values reported and used. We have updated the mooring and drag-embedment anchor cost coefficients to values from an alternate source which provides more realistic values based on the ranges from literature.